# END-TO-END REINFORCEMENT LEARNING FOR TRAFFIC SIGNAL CONTROL: REAL-TIME VIDEO TO SIGNAL DECISIONS

## ABSTRACT

Efficient traffic management at urban intersections is vital for reducing congestion and improving safety. This paper presents MD3DQN, the first End-to-End novel reinforcement learning model using surveillance video for real-time traffic signal control. The model features two main components: an image reception module, capturing traffic data from cameras positioned on signal poles, and a multi-agent decision module, where each agent manages a traffic phase. These components are connected via a bridge module for seamless integration.

Our novel Entropy Attention Mechanism enhances the multi-decision turn-based traffic signal control by leveraging uncertainty and signal phase delays, leading to more optimized decisions. Results show MD3DQN improved cumulative reward by an average of 85.2% over Fixed-time 40 and 54.4% over DQN-VTP. The entropy mechanism contributed to a 41.8% improvement upon ablation study, demonstrating its impact on faster convergence and better performance.

## 1 INTRODUCTION

Traffic congestion at intersections remains a critical issue in urban areas, causing significant delays and economic losses [Fonseca & Garcia (2021); Cheng et al. (2023)]. Efficient Traffic Signal Control (TSC) is essential, especially as urbanization continues to increase [Wei et al. (2019)].

Recent advances in Reinforcement Learning (RL) have shown great promise in improving TSC systems by dynamically adjusting signal timings in response to real-time traffic conditions, often outperforming traditional approaches in simulations [Koh et al. (2020); Li et al. (2021); Noaeen et al. (2022)]. As a result, RL has become a popular research direction for intersection control, with a growing body of literature exploring its potential.

However, despite the impressive progress in RL-based TSC models, real-world deployment remains limited. One key reason is the reliance of most RL models on processed inputs such as queue lengths, which are easily obtained in simulation environments but challenging to measure accurately in real-world settings [Comert & Cetin (2021)]. Data from loop detectors or cameras often lacks precision, and additional algorithms are required to extract accurate queue length estimates from these sources [Kessler et al. (2021); Gouran et al. (2023)]. Consequently, a seamless end-to-end solution is still lacking and two crucial components are needed: robust methods for extracting queue length from diverse data sources and the integration of these inputs into existing RL models. Until this challenge is addressed, the full potential of RL-based TSC systems cannot be realized in practical deployments.

To address this challenge, our work presents the first end-to-end solution that directly utilizes real-time video from traffic cameras to train RL models for urban TSC control. Unlike previous approaches that rely on pre-processed inputs, our model operates on raw traffic data, advancing recent studies on deep RL in complex urban networks [Xu et al. (2023); Li et al. (2023b); He et al. (2023); Wu et al. (2023)]. By training on real-world traffic data, our approach bridges the gap between simulation and practical application, making the model ready for immediate deployment in real-world traffic systems.

In summary, our contributions are as follows:

- We are the first to propose an end-to-end solution integrating surveillance camera data with RL for TSC, moving beyond simulator-based features like vehicle speed and stopping time by leveraging real-world sensor data from cameras.
- We designed a robust Entropy Attention Mechanism that significantly enhances turn-based traffic signal control within the reinforcement learning framework.
- Performance evaluations demonstrate that our model excels in real-time intersection control, not only under normal conditions but also in challenging environments such as night-time, rain, and fog, where reduced visibility and vehicle mobility pose significant challenges.
- Our extensive scripting efforts enable traffic flow research in highly realistic 3D-rendered simulations within Carla Dosovitskiy et al. (2017), offering a more accurate reflection of real-world scenarios.

## 2 RELATED WORK

The integration of video surveillance into traffic management systems has been shown to enhance vehicle detection and traffic flow predictions through deep learning techniques applied to surveillance images [Dilshad et al. (2020); Hu et al. (2021)]. Research has focused on detecting vehicle density from video data for real-time signal optimization [Jamebozorg & Hami (2024)]. Additionally, sensor fusion combining video cameras with LiDAR improves vehicle localization, though the high cost of LiDAR limits its scalability [Liu et al. (2023)]. Our research leverages the widespread deployment of traffic cameras to bridge the gap between theoretical solutions and practical implementation in signal control [Luo et al. (2018)].

Real-time vehicle detection in challenging conditions such as fog or low light has benefited from models like YOLO, which is widely adopted for traffic applications [Wang et al. (2022b); Meng et al. (2023)]. Incorporating such deep learning models into traffic systems improves detection accuracy and signal timing adjustments [Patel & Ganatra (2023); Meng et al. (2023)]. Multi-stream temporal structures have further enhanced congestion detection from video, directly supporting traffic control strategies [He et al. (2023)].

While RL methods have been effective for optimizing traffic signals [Wei et al. (2021); Xu et al. (2023)], current approaches focus primarily on theoretical simulation without practical consideration of real-world sensor inputs like surveillance cameras. Our MD3QN model extends multi-agent RL concepts by incorporating real-time video data, offering more adaptive and practical traffic control solutions [Liang et al. (2019); Huang et al. (2021)]. Multi-agent RL has demonstrated great promise in improving urban traffic flow, and video input enhances decision-making capabilities further [Wang et al. (2021); Liu et al. (2023)].

Our research addresses the limitations of existing RL-based traffic control systems, which often lack real-world applicability, by integrating video data to provide a scalable and intelligent solution for practical intersection management [He et al. (2023); Wu et al. (2023)].

## 3 PRELIMINARIES

### 3.1 TRAFFIC INTERSECTION DESCRIPTION

At a typical four-legged intersection, each approach has two lanes: one for left turns and one for straight or right turns [Papageorgiou et al. (2003)]. These lanes are grouped into lane sets, which are activated during the same signal phase when there are no conflicting movements. The incoming and outgoing lanes are defined as:

$$L_{in} = \{l_W^l, l_W^{s/r}, l_E^l, l_E^{s/r}, l_N^l, l_N^{s/r}, l_S^l, l_S^{s/r}\}, \quad L_{out} = \{l_W', l_E', l_N', l_S'\}$$

where $l_W^l$ represents the west incoming left-turn lane and $l_W^{s/r}$ represents the west incoming straight/right-turn lane. Traffic movements are defined as $(l_i^{\text{type}}, l_j')$, grouping non-conflicting lane sets for signal timing adjustments.

## 3.2 Signal Phases and Action Space

The intersection operates with four distinct signal phases, each controlling traffic from different directions and movements [Chen et al. (2015)]:

Phase 0 : West-East left-turn protection,

Phase 1 : West-East straight/right-turn,

Phase 2 : North-South left-turn protection,

Phase 3 : North-South straight/right-turn.

In each phase, non-conflicting lane sets are activated, allowing traffic to flow from specific lanes. The phase activation is represented as:

$$p_k = \{(l_i^{\text{type}}, l_j') \mid a(l_i^{\text{type}}, l_j') = 1\}$$

where $p_k$ is the active phase, and $a(l_i^{\text{type}}, l_j') = 1$ indicates that the signal is green for the movement from incoming lane $l_i^{\text{type}}$ to outgoing lane $l_j'$.

For example, during Phase 0, both $l_W^l$ (west left-turn lane) and $l_E^l$ (east left-turn lane) may have green lights, while opposing movements are stopped to prevent conflicts, as shown in Figure 1.

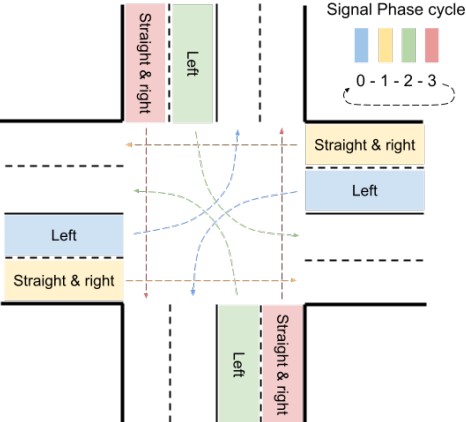

Figure 1: Traffic Intersection Signal Phases and Cycling.

Our model makes decision every 5 second, and the action space at each decision point consists of:

- *Retain*: Continue with the current phase $p_k$ for an additional 5 seconds.
- *Switch*: Transition to the next phase $p_{k+1}$, with a 3-second yellow light followed by a 5-second green light.

The action space $A$ is defined as:

$$A = \{a_{\text{retain}}, a_{\text{switch}}\}$$

where $a_{\text{retain}}$ maintains the current phase, and $a_{\text{switch}}$ initiates the transition to the next phase. This action definition ensures flexibility while simplifying decision-making. [Salah Bouktif (2021)].

## 3.3 Markov Decision Process

The traffic signal control problem is modeled as a Markov Decision Process (MDP) [Puterman (1990); Wang et al. (2023; 2022a)], defined by the tuple $(S, A, P, R, \gamma)$:

*State space*: $s_t$ includes vehicle density, queue length, and current signal phase:

$$s_t = \left(\{x(l_i), q(l_i)\}_{i \in \{W, E, N, S\}}, p_k\right)$$

*Action space*: Retaining or switching the signal phase.

*Transition function*: $P(s_{t+1} \mid s_t, a)$ models system dynamics.

*Reward function*:

$$R(s_t, a) = -\left(\alpha \sum_i t_{\text{stop}}(l_i) + \beta \sum_i q(l_i)\right)$$

where $\alpha$ and $\beta$ weigh stopping time and queue length.

The objective is to find the optimal policy:

$$\pi^* = \arg\max_\pi \mathbb{E}\left[\sum_{t=0}^{\infty} \gamma^t R(s_t, a_t)\right]$$

## 4 METHODOLOGY

### 4.1 SOLUTION OVERVIEW

Our proposed solution utilizes image-based inputs to manage traffic signal control through reinforcement learning. The images are captured at three time stamps: $t - 30$, $t - 15$, and $t_{\text{now}}$, enabling the model to capture both current and past traffic dynamics. This temporal information is crucial for allowing the model to understand evolving traffic states [Wang et al. (2020)].

The model consists of three main components:

- **Image Processing Module**: This component extracts high-level semantic features from the input images, transforming them into a lower-dimensional feature space $\Phi(I_t)$. The module is designed to be modular, allowing flexibility to replace it with various image processing techniques as required.

- **Feature Space Mapping (Bridge)**: This part of the model serves as a bridge, aligning and mapping the extracted image features with additional traffic metrics, such as vehicle density and queue lengths, into a unified decision-making space. The mapping is represented by $f : \mathbb{R}^{H' \times W' \times D} \to \mathcal{S}_{\text{bridge}}$, where $\mathcal{S}_{\text{bridge}}$ is the intermediate space for reinforcement learning.

- **Multi-Decision Agent**: Each agent focuses on one lane set, processing its features through an entropy module. The entropy module incorporates both phase timing (phase timing distance, i.e., the time until the next or subsequent signal phases) and state uncertainty. The agent optimizes a policy $\pi(s_t)$, balancing rewards such as minimizing queue lengths and reducing vehicle stopping times.

This modular and flexible architecture enables the system to make informed, real-time traffic control decisions based on image data and traffic metrics. As shown in Figure 2, the model integrates these components seamlessly to improve traffic flow and decision-making efficiency.

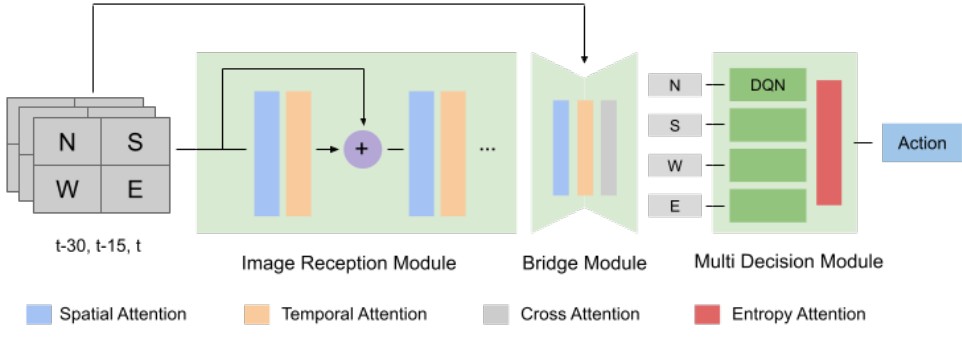

Figure 2: Overall MD3DQN Model Structure.

## 4.2 INPUT REPRESENTATION

The input to the model consists of images captured at three distinct time stamps: $t - 30$, $t - 15$, and $t_{\text{now}}$, where $t_{\text{now}}$ represents the current time. Each time stamp includes four directional images representing North (N), South (S), West (W), and East (E) orientations of the intersection. Thus, a total of 12 images are fed into the model as input, capturing the real-time and historical traffic conditions from surveillance cameras positioned at the intersection.

This temporal sequence of images provides the model with both current and past traffic states, enabling it to learn traffic dynamics over time. These inputs, combined with temporal features, are processed by the model to inform the decision-making process. The input images are visualized in Figure 3.

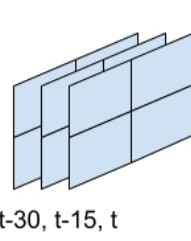 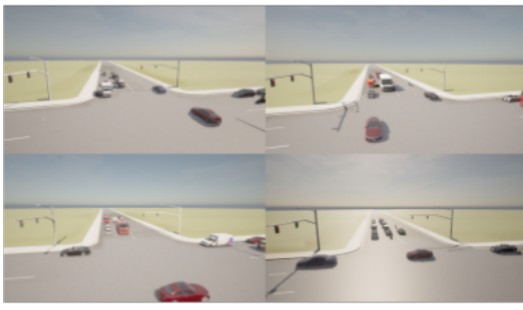

t-30, t-15, t

Figure 3: Model Input: Traffic Images from Different Directions at Three Time Stamps.

## 4.3 VIDEO RECEPTION MODULE

The video reception module processes traffic images captured at three timestamps: $t-30$, $t-15$, and $t_{\text{now}}$, covering four directions (North, South, West, and East). This module extracts structured features from raw images, which are then used for reinforcement learning. We tested three approaches for feature extraction:

1. **Customized ResNet**: This method applies spatial and temporal attention [Vaswani et al. (2017)] for a ResNet [He et al. (2016)]. Spatial attention $\alpha_{\text{sp}}(p)$ highlights key regions in each image:

$$\alpha_{\text{sp}}(p) = \frac{\exp(W_{\text{sp}}^T p + b_{\text{sp}})}{\sum_{p'} \exp(W_{\text{sp}}^T p' + b_{\text{sp}})}, \quad \alpha_{\text{tmp}}(t_i) = \frac{\exp(U_{\text{tmp}}^T h_{t_i} + c_{\text{tmp}})}{\sum_{t_j} \exp(U_{\text{tmp}}^T h_{t_j} + c_{\text{tmp}})}$$

where $W_{\text{sp}}$, $U_{\text{tmp}}$, and corresponding biases are learnable parameters. Temporal attention captures traffic flow changes across timestamps. Both attentions are combined with residual connections $R(x) = x + \text{Attention}(x)$ to retain key information.

2. **YOLO**: YOLOv10-S Pre-trained model [Ao Wang et al. (2024)], a real-time object detection model, was fine-tuned to detect vehicles, positions, and densities.

3. **ViT**: The ViT-Base-Patch16-224 model [Wu et al. (2020)], utilizing self-attention, captures long-range image dependencies and extracts traffic features like congestion and vehicle distribution.

These approaches enable robust feature extraction, feeding crucial information into downstream RL tasks to optimize traffic signal control.

## 4.4 BRIDGE LAYER MODULE

The Bridge Layer Module maps the enriched feature space from the Video Reception Module into actionable inputs for the SUMO pre-trained RL agent [Lopez et al. (2018)]. The concatenated feature map $F$, combining original input and attention-enhanced features, is transformed into a lower-dimensional space for decision-making via:

$$\tilde{F} = \phi(W_b F + b_b)$$

where $\phi$ is a non-linear activation, $W_b$ the weight matrix, and $b_b$ the bias vector. The output $\tilde{F}$ captures both spatial and temporal relationships.

A cross-attention mechanism $\alpha_{\text{cr}}$ [Cai & Wei (2020)] integrates spatial focus and temporal changes:

$$\alpha_{\text{cr}}(F_{\text{sp},i}, F_{\text{tmp},j}) = \frac{\exp(W_c^T(F_{\text{sp},i} \oplus F_{\text{tmp},j}))}{\sum_k \exp(W_c^T(F_{\text{sp},i} \oplus F_{\text{tmp},k}))}$$

where $F_{\text{sp},i}$ and $F_{\text{tmp},j}$ represent spatial and temporal features, respectively. This mechanism ensures effective integration of spatial and temporal dependencies. The final output is sent to the Multi-Decision Module for traffic signal control.

### 4.5 MULTI MINI DECISION AGENT MODULE

The Multi Mini Decision Agent Module consists of four mini-agents [Zhang et al. (2021); Hernandez-Leal et al. (2020)], each responsible for one lane set (North-South left, North-South straight/right, West-East left, West-East straight/right), processing features through an entropy module that accounts for both phase timing and state uncertainty.

Each mini-agent calculates an action score $S_i = \sigma(W_d F_i + b_d)$, where $\sigma$ is the sigmoid function, $W_d$ and $b_d$ are the weight matrix and bias, and $F_i$ represents the feature set for a given lane. The scores $S_1, S_2, S_3, S_4$ are then normalized into probabilities:

$$P_i = \frac{S_i}{\sum_{j=1}^4 S_j}$$

Shannon entropy [Lin (1991)] $H = -\sum_{i=1}^4 P_i \log(\max(P_i, \epsilon))$, is used to quantify decision uncertainty and adjust the decay constant $k' = k \cdot H$. The weights $w_i = e^{-k' d_i}$ are then computed based on phase timing distance $d_i$. These integrated weights are multiplied by the positional embedding [Vaswani et al. (2017); Li et al. (2023c)] of the traffic signal phase sequence delay, ensuring that phase timing is incorporated into the decision-making process for more robustness:

$$w_i' = w_i \cdot \alpha_{\text{cr}}(S_i)$$

Finally, the weighted scores are used to compute the Q-values for two possible actions, with the action corresponding to the larger Q-value being selected:

$$Q_{\text{retain}}, Q_{\text{switch}} = f_Q(w_i' \cdot S_i)$$

The Entropy Attention Module, as shown in Figure 4, boosts performance, with its effectiveness proven by the ablation study results.

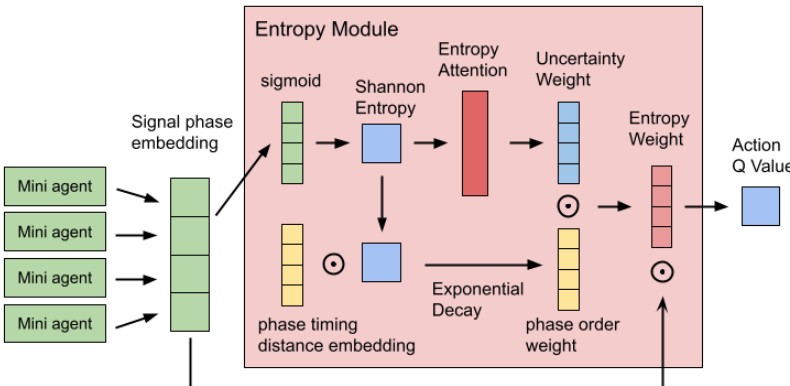

Figure 4: Entropy Attention Module: Integrating phase timing distance, entropy, and uncertainty weight to compute entropy-adjusted weights for each decision.

### 4.6 DUELING NETWORK

The decision-making process in the Multi Mini Decision Agent Module uses a Dueling Network architecture [Wang et al. (2016)]. The Q-value $Q(s, a)$ is split into the value function $V(s)$, which

estimates the reward of being in a state, and the advantage function $A(s, a)$, which measures the benefit of an action:

$$Q(s, a) = V(s) + \left( A(s, a) - \frac{1}{|A|} \sum_{a'} A(s, a') \right)$$

This separation improves learning by distinguishing the value of the state from the relative advantage of each action, stabilizing the decision-making process [Konda & Tsitsiklis (1999)].

The Dueling Network, combined with entropy-based attention and the multi-agent framework, enhances the system's ability to make intelligent, real-time traffic control decisions.

# 5 EXPERIMENT

## 5.1 MULTI MINI DECISION AGENT PRE-TRAINING ON SUMO

The Multi Mini Decision Agent Module was pre-trained in the SUMO traffic simulation environment using non-image features such as vehicle counts, queue lengths, and cumulative stopping times. The objective was to establish decision-making capability in traffic signal control before introducing complex video inputs.

The intersection settings mirrored those in later stages, with 3 timestamps ($t - 30$, $t - 15$, and $t_{\text{now}}$) per lane set. Each timestamp captured cumulative stopping time $t_{\text{stop}}(l_i)$ and queue length $q(l_i)$ for lane set $l_i$. This simplified pre-training setup ensures faster convergence and explainability before integrating video-based inputs.

## 5.2 CARLA FINE-TUNING

Fine-tuning in the Carla simulation environment involved using real-time image inputs to capture dynamic traffic flow. Carla offers more realistic vehicle dynamics, making it crucial for testing in real-world intersection scenarios. The setup and data collection were similar to SUMO, with images taken at $t - 30$, $t - 15$, and $t_{\text{now}}$.

A key component is the weighted reward function [Li et al. (2023a); Peters et al. (2010)]:

$$R_{\text{weighted}} = 0.4 \cdot r_1 + 0.3 \cdot r_2 + 0.2 \cdot r_3 + 0.1 \cdot r_4$$

This reflects the cumulative impact of decisions, with rewards normalized across light ($p_{\text{light}} = 0.01$) and heavy ($p_{\text{heavy}} = 0.04$) traffic flows. Stopping time weight $\alpha = 0.1$ and queue length weight $\beta = 1$ were chosen to balance both metrics based on prior training.

Fine-tuning showed that the MD3DQN model can handle complex, real-world scenarios by leveraging both image inputs and weighted rewards, optimizing signal control in adaptive and intelligent ways.

## 5.3 HYBRID ONLINE-OFFLINE TRAINING STRATEGY

We use a hybrid online-offline strategy to improve training efficiency [Nair et al. (2020)]. Let $\mathcal{E}$ be the Carla environment and $\pi_\theta(a|s)$ the policy parameterized by $\theta$.

**1. Data Collection** We interact with $\mathcal{E}$ to gather experiences $\mathcal{D}_0 = \{(s_t, a_t, r_t, s_{t+1})\}$, storing them in a buffer $\mathcal{B}$: $\mathcal{B} = \mathcal{D}_0$.

**2. Offline Training** In the offline phase, $\pi_\theta$ is updated using mini-batches of size 150 from $\mathcal{B}$. The policy parameters are adjusted as $\theta \leftarrow \theta - \eta \nabla_\theta \mathcal{L}(\theta)$, where $\eta = 0.0005$. We run 5 iterations per cycle, improving sample efficiency.

**3. Periodic Online Updates** After each offline cycle, new experiences $\mathcal{D}_n$ are collected from $\mathcal{E}$ and added to $\mathcal{B}$, replacing old data: $\mathcal{B} \leftarrow \mathcal{B} \cup \mathcal{D}_n$. Roughly 10-20% of the buffer is updated.

**4. Iteration** This process repeats: $\pi_\theta^{(n)} \xrightarrow{\text{Offline}} \pi_\theta^{(n+1)} \xrightarrow{\text{Online}} \mathcal{D}_{n+1} \rightarrow \mathcal{B}$.

**Hyperparameters** Key parameters include: buffer size = 2000, $\gamma = 0.97$, $\eta = 0.0005$, $\tau = 0.01$, batch size = 150, and $\epsilon$ annealing from 1 to 0.1 over 20,250 steps.

## 5.4 METRICS

We evaluated the MD3DQN model using two key metrics [Kim et al. (2023); Ault & Sharon (2021)]: average stopping time (AST) and average queue length (AQL) [Akçelik (1980)], measured over a single run in both light ($p_{\text{light}} = 0.01$) and heavy ($p_{\text{heavy}} = 0.04$) traffic conditions. The metrics were balanced to ensure equal importance for both traffic flows, accounting for the difference in probabilities.

AST is defined as:

$$\text{AST} = \frac{1}{T} \sum_{i=1}^{N} t_{\text{stop}}(S_i), \quad \text{AQL} = \frac{1}{T} \sum_{i=1}^{N} q(S_i)$$

where $t_{\text{stop}}(S_i)$ and $q(S_i)$ represent the stopping time and queue length for lane set $S_i$, respectively, and $T$ is the flow duration in minutes. These metrics provide a comprehensive view of the traffic dynamics in both light and heavy traffic conditions.

## 5.5 BASELINES

Since no direct comparisons exist for our end-to-end video-to-signal control model, we selected two fixed-time signal control methods and a customized model for baseline comparison [Ouyang et al. (2024); Zhao et al. (2021)]. Additionally, we tested our MD3DQN-Res model under extreme conditions (fog, rain, and night) to evaluate its generalization without explicitly training on those scenarios.

The baseline methods are:

1. *Fixed-Time 30*: A signal timing scheme where each phase is set to 30 seconds, independent of traffic conditions.

2. *Fixed-Time 40*: Similar to Fixed-Time 30, but with each phase lasting 40 seconds.

3. *DQN-VTP*: A customized model combining a YOLO-based vehicle detection system to assess traffic pressure (the number of vehicles approaching the intersection on specified lane sets). These features are then used to train a DQN model to control traffic signals via reinforcement learning.

The traffic pressure detection system is illustrated in Figure 5, which shows how traffic pressure is estimated for use in the reinforcement learning model.

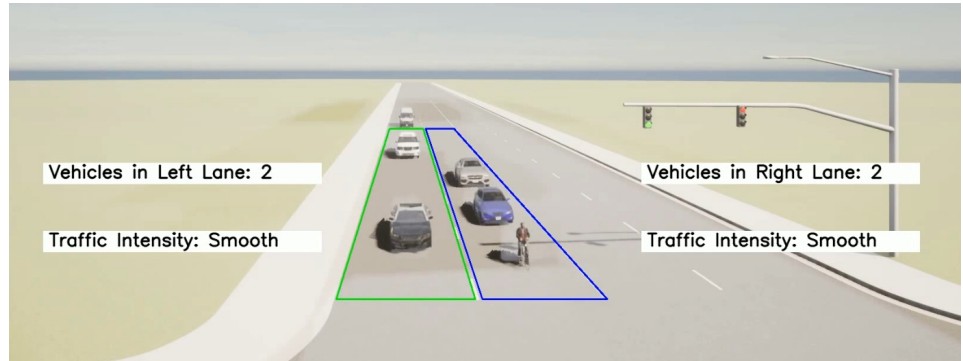

Figure 5: Traffic Pressure Detection System on predefined lane sets.

We further tested the MD3DQN-res model under extreme weather conditions (fog, rain, and night), without explicit training for these scenarios, to assess its generalization capabilities. As shown in Figure 6, the model was tested under various weather conditions to demonstrate its adaptability.

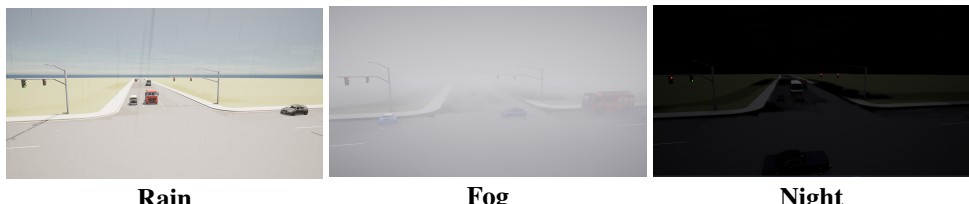

|  |  |  |
| :---: | :---: | :---: |
| **Rain** | **Fog** | **Night** |

Figure 6: Camera shot from different extreme weather conditions.

## 5.6 RESULTS

The MD3DQN variants were evaluated against several baselines, as shown in Table 1. The MD3DQN-res model achieved an average reward (AR) of -13.76, a 54.4% improvement over DQN-VTP (-30.19). MD3DQN-yolo and MD3DQN-ViT also outperformed DQN-VTP with AR values of -12.46 and -18.23, respectively.

For average stopping time (AST), MD3DQN-res reduced vehicle stopping time by 57.7%, reaching 267.72 seconds per minute compared to 633.06 seconds for DQN-VTP. Similarly, average queue length (AQL) saw a 50.8% reduction, with 27.94 vehicles per minute versus 56.74 for DQN-VTP.

As shown in Figure 7, MD3DQN consistently achieved better rewards and lower stopping times across varying conditions. Even in challenging weather like fog, rain, and night, MD3DQN-res outperformed DQN-VTP baseline without needing retraining.

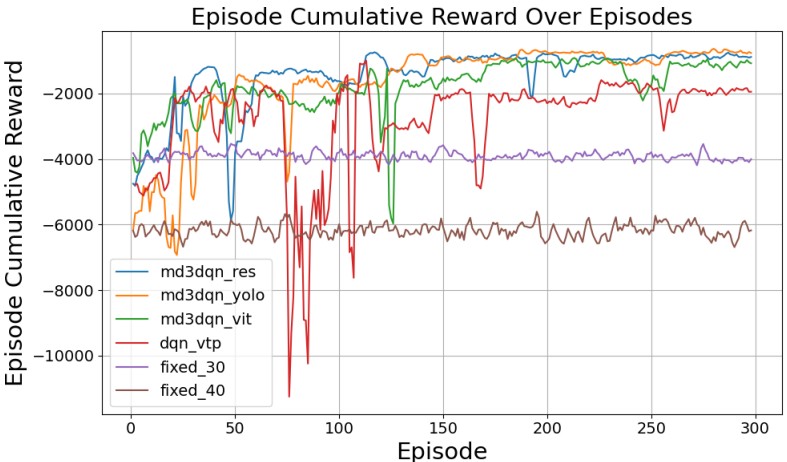

Figure 7: Cumulative Reward Comparison: MD3DQN vs. Baseline Models.

Overall, MD3DQN demonstrated superior stability and efficiency, making it an effective solution for real-time traffic signal control.

## 5.7 ABLATION STUDY: IMPACT OF ENTROPY ATTENTION MECHANISM

The entropy attention mechanism provides a significant boost to the model's performance by dynamically adjusting decisions based on signal phase sequence delays and state uncertainty. As shown in Figure 8, MD3DQN-res with the entropy mechanism achieved an average reward (AR) of -38.97 under rain conditions, whereas the model without entropy (MD3DQN-res_no_entropy) only reached -65.57 under fog conditions. This represents an improvement of approximately 40.6%, demonstrating the importance of entropy in both faster convergence and enhanced overall performance.

By considering the sequence delay between signal phases, the entropy module allows the model to make more informed decisions, leading to more efficient and adaptive traffic management, even under challenging weather conditions.

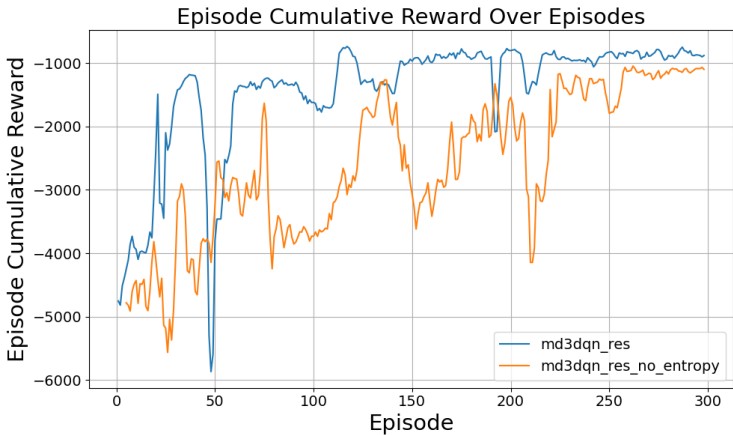

Figure 8: Impact of Entropy Attention on Average Reward.

## 6 CONCLUSION

This paper introduced MD3DQN, the first End-to-End video-to-signal control framework for urban intersections, integrating real-time video inputs with reinforcement learning. MD3DQN surpasses traditional baselines, including fixed-time and object-detection-based DQN approaches, in metrics such as average stopping time (AST), queue length (AQL), and average reward (AR), as shown in Table 1.

As shown in Figure 8, the ablation study of the entropy-based attention mechanism highlights its key role in enhancing decision accuracy by accounting for signal phase timing and uncertainty.

Our model also demonstrated robust performance in extreme weather conditions like rain, fog, and night, maintaining superior performance compared to DQN-VTP without the need for retraining, as shown in Table 2.

Future work will explore multi-intersection control, pedestrian integration, and further refinement of the model in complex environments to enhance real-time traffic management.

| Method | AR (/min) ↑ | AST (s/min) ↓ | AQL (veh/min) ↓ |
|---|---|---|---|
| MD3DQN-res | -13.76 | 267.72 | 27.94 |
| MD3DQN-yolo | -12.46 | 238.04 | 25.72 |
| MD3DQN-ViT | -18.23 | 346.52 | 37.73 |
| DQN-VTP | -30.19 | 633.06 | 56.74 |
| Fixed-time 30 | -59.03 | 1790.99 | 57.82 |
| Fixed-time 40 | -93.00 | 3007.91 | 73.84 |

Table 1: Performance result of MD3DQN against other baseline methods. AR (Average Reward), AST (Average Stopping Time), AQL (Average Queue Length).

| Weather | Model | AR (/min) ↑ | AST (s/min) ↓ | AQL (veh/min) ↓ |
|---|---|---|---|---|
| Rain | MD3DQN-res | -38.97 | 1450.76 | 117.40 |
| | DQN-VTP | -49.42 | 1927.92 | 119.44 |
| Fog | MD3DQN-res | -65.57 | 2213.90 | 237.05 |
| | DQN-VTP | -69.44 | 2577.51 | 244.65 |
| Night | MD3DQN-res | -86.74 | 2776.76 | 262.08 |
| | DQN-VTP | -101.94 | 2598.96 | 288.22 |

Table 2: Comparison of models under extreme weather conditions.

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
