# OpenReview forum: "End-to-End Reinforcement Learning for Traffic Signal Control: Real-Time Video to Signal Decisions"
_ICLR.cc/2025/Conference — ICLR 2025 Conference Withdrawn Submission_

### Official Review · Reviewer_Wf32 · 2024-10-31

**Soundness:** 2
**Presentation:** 3
**Contribution:** 2
**Rating:** 3
**Confidence:** 5

**Summary:**

This paper presents MD3DQN, an end-to-end reinforcement learning model for real-time traffic signal control using surveillance video. The model integrates video-based traffic data with a multi-agent decision system to dynamically adjust traffic signals at intersections. A novel Entropy Attention Mechanism enhances decision-making by handling phase timing delays and uncertainties.

**Strengths:**

1. This paper proposes a novel method for traffic signal control based on visual images.
2. Different from existing methods based on road statistics, this method uses information similar to camera images as input, which can transition to real scenes more smoothly.
3. The proposed multi-agent decision-making algorithm based on Entropy Attention Mechanism can improve the control performance.

**Weaknesses:**

1. The experiment is not sufficient: the authors only compared a baseline method on one dataset.
2. Ablation experiments are not perfect: the authors only verified the effect of Entropy Attention. For the proposed visual processing module, the authors should also conduct ablation experiments to illustrate its advantages over existing non-visual solutions.
3. There is still a gap between the superiority of the proposed method and the description in the paper: in real applications, we need to pay attention to the real-time nature of decision-making and the computational and storage cost of the control method. The proposed method uses a visual module and multi-agent decision-making, which introduces additional computational time and storage cost, which the author did not report in the paper.

**Questions:**

1. Existing methods based on statistical information such as pressure and other indicators are still easy to deploy in real scenarios using roadside sensors. What are the significant advantages of the proposed vision-based method?
2. Why is the green light duration set to 5 seconds? This seems too short in practical applications.
3. The method used as a comparison is too simple. Since the proposed method also uses road statistics as part of the input, the authors should consider comparing it with the existing SOTA methods.
4. Since the input information contains statistical information of visual images and roads, the authors should conduct ablation experiments to verify the effectiveness of the proposed vision-based method.
5. The authors tested rainy and foggy scenes, but this seems to only illustrate the effectiveness of the existing visual processing module used. The authors should consider using more datasets with different traffic flow characteristics to illustrate the generalization and robustness of the proposed method.
6. The authors only considered the traffic scenario of a single intersection and did not report the computational and storage overhead of the method, so the scalability of this method needs further consideration.

---

### Official Review · Reviewer_4Lc7 · 2024-10-31

**Soundness:** 3
**Presentation:** 2
**Contribution:** 3
**Rating:** 3
**Confidence:** 4

**Summary:**

This paper tries to solve traffic signal control problems using an end-to-end method with raw images instead of well-processed structured data. It adopts classic video reception models to extract readable features from raw images. It also proposes a bridge module and a Dueling Network-based multi-mini decision agent module with entropy attention for decision-making based on the preprocessed features. This work novelly addresses TSC using “real-world” sensor data (images from Carla simulator), reducing the sim-to-real gap. Experiments on a simple signalized intersection with different weather settings are conducted to verify the feasibility of the proposed method.

**Strengths:**

* The originality is good. This paper tries to solve a new problem about the sim-to-real gap by addressing TSC from raw camera-captured images.

* The clarity is good. It explains the proposed methods well somehow.

**Weaknesses:**

* More related work for current TSC methods would be better, such as problem formulations and conventional or RL-based TSC methods.
* The quality/definition of figures shown in this paper could be greatly improved.
* It’s not clear how you extract temporal and spatial dependencies separately.
* All results are based on one single run, which might lack confidence for uncertainty evaluation. Experimental results based on multiple random seeds should be presented to be more convincing.
* How do you get the following conclusion from Figure 8: “MD3DQN-res with the entropy mechanism achieved an average reward (AR) of -38.97 under rain conditions, whereas the model without entropy (MD3DQN-res no entropy) only reached -65.57 under fog conditions.”
* It lacks reasonable non-RL or RL baseline methods, such as Max Pressure.
* The experiments seem to be based on a very simple four-approach single-intersection control task. More complex scenarios could be included such as single intersections with different structures and phase settings and multi-intersection control tasks.
* Besides, there is no detailed introduction about the task settings, such as traffic flow setting, episode horizon, etc. The experiments seem to be not convincing.
* All raw pictures are not from real-world sensors but from a 3-D simulator Carla. This can be improved but not a major concern.
* Implementation code should be provided.
* The writing and organization should be further improved.

**Questions:**

* What’s the sequence phase delay? It’s not clearly explained.
* Definitions for $r_{1,2,3,4} $ in the weighted reward function are not explained.

---

### Official Review · Reviewer_A6Ns · 2024-11-02

**Soundness:** 1
**Presentation:** 2
**Contribution:** 1
**Rating:** 1
**Confidence:** 4

**Summary:**

In this paper, an end-to-end traffic signal control framework is proposed leveraing end-to-end perception with RL learning. An entropy attention mechanism is further developed. The exepriment results demonstrate better convergence in reward learning.

**Strengths:**

A novel end-to-end traffic signal control pipeline using RL and entropy attention mechanism

**Weaknesses:**

Insufficient evaluations:

1) The main results rely solely on a single reward curve. To improve reliability, multiple trials should be conducted with range and upper-lower bounds included.

2) The testing scenario lacks representativeness. Evaluations under various real-world scenarios, different traffic densities, and real-world datasets are needed. Without this, the end-to-end setup is not adequately validated.

3) Limited evaluation metrics. Using only a reward function is not an objective performance measure. Table 2 suggests significant variation in performance under different weather conditions, raising concerns about the framework’s stability.

Concerns in the problem setup: The control signals should ideally be treated as a unified system across multiple intersections or road networks. However, the proposed method is limited to a single intersection, leaving generalizability unaddressed.

Concerns regarding framework novelty: All components (e.g., advantage DQN, attention mechanisms) have been previously proposed, offering little incremental innovation.

**Questions:**

See "Weaknesses"

---

### Official Review · Reviewer_jXP3 · 2024-11-02

**Soundness:** 2
**Presentation:** 2
**Contribution:** 2
**Rating:** 3
**Confidence:** 3

**Summary:**

The paper presents an integrated approach to traffic signal control by leveraging computer vision for real-time traffic monitoring and reinforcement learning (RL) for adaptive signal management. The proposed system captures traffic conditions through video feeds, employing computer vision techniques to process images and extract relevant features. Based on this information, the RL model is trained to manage traffic light phases, allowing for dynamic adjustments to improve traffic flow.

The key contributions of the paper include the development of an end-to-end traffic signal control framework that combines image processing with RL decision-making, aiming to optimize signal timings based on real-time data. The authors illustrate the potential for this integrated approach to enhance traffic management systems, thereby contributing to the ongoing advancements in intelligent transportation systems. The experimental results, although limited in scope, demonstrate the feasibility of applying this method in controlled environments, suggesting a foundation for future research and refinement in traffic signal control applications.

**Strengths:**

1. The integration of computer vision with reinforcement learning for traffic signal control is a novel approach, highlighting the potential for enhanced real-time decision-making.

2. The use of real-time video feeds demonstrates a practical application of existing computer vision techniques, aligning the research with real-world traffic management needs.

3. The paper clearly defines the problem and objectives, presenting a well-structured framework that facilitates understanding of the proposed method.

4. The research addresses critical issues in urban mobility, contributing to efforts in intelligent transportation systems, and lays the groundwork for future enhancements in traffic signal control strategies.

**Weaknesses:**

1. Lack of Novelty in Model Design: While the integration of computer vision and reinforcement learning is a promising approach, the model design appears simplistic. The system utilizes only four signal phases and two actions (change or maintain the current state), which may limit its effectiveness in handling the complexities of real-world traffic scenarios. Enhancing the model to incorporate more signal phases and actions could improve its adaptability and performance.

2. Limited Experimental Scope: The experimental validation is conducted in small-scale scenarios, which may not adequately represent the challenges faced in larger, more complex urban environments. Expanding the experiments to include larger and more varied traffic conditions would provide a better assessment of the system’s effectiveness and robustness.

3. Absence of Comprehensive Baselines:The paper does not compare the proposed system with a wide range of established baselines, particularly recent RL-based traffic signal control methods. Notably, including comparisons with methods like FRAP/MPLight approaches would strengthen the evaluation and demonstrate the relative performance of the proposed system.

4. Insufficient Justification for Methodology: The authors do not adequately justify their choice of methodologies or explain how the proposed system improves upon existing techniques. Providing more detailed discussions on the advantages of their approach compared to previous works in the field would enhance the paper's credibility.

5. Limited Use of Advanced Simulation Tools: The experiments rely solely on SUMO for simulation, which, while useful, may not capture the full complexity of real-world traffic conditions. Utilizing more advanced or contemporary simulators could offer richer insights into the system’s performance and scalability.

**Questions:**

Please refer to the Weakness part.

---

### Official Review · Reviewer_WaZr · 2024-11-10

**Soundness:** 2
**Presentation:** 2
**Contribution:** 2
**Rating:** 3
**Confidence:** 4

**Summary:**

This paper introduces MD3DQN that aims to optimize traffic light control via deep reinforcement learning. MD3DQN is able to intake raw images from surveillance cameras and output a decision on whether the traffic light should advance to the next state or stay at the current state. The optimization goal is to minimize vehicle stop time and queuing length. The authors innovated the Entropy Attention Mechanism that enhances signal control by factoring in uncertainty and temporal information, which reportedly largely improves decision performance especially under challenging weathers. The author claim MD3DQN achieves lower average stopping time and queue length compared to a fixed-time scheduled traffic light switch and a DQN baseline.

**Strengths:**

Applying reinforcement learning for traffic light control is novel. The problem formulation is very clear.

**Weaknesses:**

- Well the paper aims to build an end-to-end system to produce decisions directly from sensor inputs, the author didn't explicitly state the benefit as compared to a modulized system that first estimate traffic density (such as number of cars waiting) and decide based on that. At line 042-043, the author believes an end-to-end solution is more robust but I wonder what additional information can be passed to the RL system? From the weather conditioned experiments I can infer that the additional information might be the visibility induced uncertainty, and your RL system is able to consume this information at decision making. What if such information is provided in a more direct way? For example a visibility score from the image.
- The methodologies needs more detail, such as what are the image features from the video perception module, a hidden vector or a BEV feature map?
- The paper needs more clarification on the methods (more on the questions below).

**Questions:**

- Could you clarify your definition of end to end? Do the gradients in the decision module flow back into the image processor?
- In Section 4.3, the relation between "ResNet", "YoLo", "ViT" is unclear: are these separate methods experimented in your pipeline? If yes, do you produce the same format of image features that are passed to the `bridge`? You mentioned ResNet gives you $\alpha_{sp}$ and $\alpha_{tmp}$, what about YoLo and ViT? And how does  $\alpha_{sp}$ and $\alpha_{tmp}$ relate to  $F_{sp}$ and $F_{tmp}$ in line 274?
- Does the subscription `j` refer to another lane set as `i`? Please clarify.
- In line 194, is the `phase timing distance` the same for all agents at the decision time since you make decision for all agents together? Because at line 145, you mentioned that the model decides every 5 seconds, so is this `phase timing distance` just 5?
- In line 301, could you clarify how you designed $f_Q$?
- Line section 5.1, when you pretrain without images, do you still apply the bridge module and the entropy attention mechanism? They seem to be image based.
- How is the DQN-VTP baseline differ from yours?

---

### Note · Authors · 2024-11-25

I have read and agree with the venue's withdrawal policy on behalf of myself and my co-authors.